# Changes in the Properties in Bimodal Mg Alloy Bars Obtained for Various Deformation Patterns in the RSR Rolling Process

**DOI:** 10.3390/ma15030954

**Published:** 2022-01-26

**Authors:** Andrzej Stefanik, Piotr Szota, Sebastian Mróz, Marcin Wachowski

**Affiliations:** 1Faculty of Production Engineering and Materials Technology, Czestochowa University of Technology, 42-201 Częstochowa, Poland; piotr.szota@wip.pcz.pl (P.S.); sebastian.mroz@pcz.pl (S.M.); 2Faculty of Mechanical Engineering, Military University of Technology, 00-908 Warsaw, Poland; marcin.wachowski@wat.edu.pl

**Keywords:** radial shear rolling, AZ31 magnesium alloys, bimodal-structure bars, FEM

## Abstract

The paper presents the theoretical and experimental research conducted to date regarding the possibility of obtaining round bars from AZ31 magnesium alloy with a bimodal structure rolled in the radial shear rolling process (RSR) technology. There is no analysis of the impact of the deformation path (distribution of deformation in individual passes) on the mechanical properties and the obtained bar structure. The feedstock, namely, AZ31 magnesium alloy round bars with a diameter of 30 mm, were rolled in RSR to the final diameter of 15 mm with different levels of deformation in successive passes, at a temperature of 400 °C. The bars obtained as a result of the RSR rolling process have different hardness on the cross-section as well as a characteristic gradient grain size distribution. Based on the conducted research, it can be concluded that the use of a larger number of passes with a smaller cross-section reduction will result in an improved formation of a bimodal structure consisting of a highly fragmented near-surface structure and in the half of the radius of the structure of fragmented grains at the boundaries of larger grains.

## 1. Introduction

The intensive development of technologies that has been observed for over a dozen years, facilitating the production of materials and finished products where the properties are shaped not only by changes in the chemical composition or heat treatment but primarily by means of unique methods of plastic processing, has led to the popularization of severe plastic deformation (SPD) methods [1,2,3]. They make it possible to obtain materials with a fine-grained microstructure, often ultra-fine grained (UFG), which leads to an increase in the strength of products obtained in the SPD process. With their highly fragmented structure obtained in production, the materials are characterized by a large number of grain boundaries, strongly affecting their mechanical and physical properties. The currently used SPD methods allow producing materials with high structural isotropy without disturbing the continuity of the internal structure [4,5,6]. However, with the simultaneous increase of the strength properties in fine-grained materials, a decrease in plastic properties is inevitable. Production of materials (products) may thus be a solution, e.g., materials in the form of bars with a bimodal structure, i.e., those in which the structure change takes place using a gradient progression from larger grains (in the central part of the bar) and decreasing in size towards the surface of the bar, where an ultra-fine-grained structure will develop [7,8]. Many known SPD methods cannot be directly applied to produce a finished product of appropriate dimensions as a construction material. One of the most critical reasons is their energy consumption preventing industrial application, which, due to the requirements to reduce energy consumption because of global warming, disqualifies them as construction materials. Currently, the most widely used methods to produce materials with a bimodal or ultra-fine grain structure on an industrial scale are accumulative roll bonding (ARB) for sheet metal production [9,10], high-pressure torsion extrusion (HPTE) [11,12], KoBo extrusion with a moving die [13,14] and radial shear rolling (RSR) [15,16].

The radial shear rolling (RSR) process is used for rolling round bars made of various metallic materials, including steel [17,18], titanium alloys [19,20], magnesium alloys [21,22], aluminum alloys [23,24] and zirconium alloys [25]. This process was developed at MISIS (National University of Science and Technology, Moscow, Russia) initially as a tube-forming process, and then adapted to produce round bars [26,27,28]. The principle of the RSR process is the characteristic rotational trajectory of the deformed metal, which causes the development of intense tangential stresses. It is possible by introducing angular cylinders inclined at an angle, causing both circumferential deformations and as axial torsion of the deformed band [26]. During RSR, in the roll gap, the stress state diagram is close to the all-around compression with additional shear stress resulting from forced metal flow along a helix varying along the radius from the axis of the rolled bar. The diagram of the RSR rolling process and the flow of the deformed metal is presented in the paper [25,26,29].

Naizabekov et al. [29] showed that RSR allows the obtaining of a bimodal structure with different grain size and morphology, strongly fragmented and equiaxial in the near-surface layers and larger elongated in the rolling direction similar to the grains obtained in the classic multi-pass rolling process and austenitic steel bars with improved mechanical properties while maintaining adequate ductility. On the other hand, Akopyan et al. [30] analyzed the formation of the gradient structure in bars made of the Al-Zn-Mg-Fe-Ni aluminum alloy when using RSR. The resulting products were bars with a gradient structure with an ultra-fine-grained surface layer and a coarse-grained core, which caused a reduction of the microhardness to 105 HV in the near-surface zone and an increase in microhardness to 145 HV in the central zone of the rolled bars.

Magnesium alloys are some of the prospective light metals very popular in the industry. These alloys are widely used in research on obtaining materials with a UFG or bimodal structure by a variety of SPD methods, among which the RSR is widely used. Dobatkin et al. presents research results on structural changes, texture and mechanical properties of AZ41 alloy rolled on a multi-pass basis using RSR [31]. The authors evidenced that the use of the RSR process allowed the obtaining of a structure characterized by an average grain size ranging from 1.5 to 3.5 µm and prismatic texture. Similar results of the structure analysis for RSR-rolled AZ31 alloy bars were obtained in [32], where the texture determined for the rolled bars was comparable to the textures obtained in the twist-extrusion (TE) process. On the other hand, the authors presented in earlier works [21] the results of comparative tests of the properties of AZ31 alloy bars obtained in the classic rolling process in modified grooves with bars produced in the RSR process. The above-cited papers show that with fewer passes in the RSR process, it is possible to produce bars with better mechanical properties, characterized by narrower dimensional tolerances in cross-sections compared to the classic rolling process. Theoretical and experimental research results on the effect of the rolling speed on the bar torsion and the possibility of obtaining a bimodal structure in RSR processing of AZ31 alloy bars, has been presented by the authors in [33]. The authors showed that increasing the rolling speed is advantageous due to increasing shear stresses responsible for activating additional deformation mechanisms and grain refinement. Furthermore, the authors evidenced that bars with a different structure were produced in the process. In the axial zone of the bars, the grain size was larger (about 35 µm) than in the near-surface zones of the bar (about 6 µm).

In the theoretical and experimental research conducted to date on obtaining bars with a bimodal structure from magnesium alloys, rolled in the RSR technology, no analysis is present of the impact of the deformation path (distribution of deformation in individual passes) on the mechanical properties and structure of bars thus obtained. Thus, the goal of the paper was to determine the effect of the deformation path on the properties and structure of the obtained bars by applying various rolling patterns. For this purpose, numerical simulations and experimental tests were applied to the RSR process in bars with a diameter of 15 mm from a batch with an initial diameter of 30 mm made of AZ31 magnesium alloy. Three variants of deformation pattern tests were carried out: I: six passes-small deformation in a single pass, II: four passes-medium deformation in a single pass and III: three passes-large deformation in a single pass. Strength, microhardness and microstructure of bars and the material in a non-deformed state were tested, which allowed determining of the impact of the applied deformation pattern on the bimodality of the bars.

## 2. Materials and Methods

In the RSR tests, the magnesium alloy AZ31 for plastic processing was used, with the chemical composition defined by the manufacturer specified in Table 1.

The accuracy of numerical modeling depends largely on the accurate determination of the properties of materials used for simulations. A model of the deformed medium developed by the authors and determined from plastometric tests of compression tests performed using the Gleblee 3800 metallurgical process simulator (Dynamic Systems Inc., Poestenkill, NY, USA) was used for numerical testing. The plastometric tests for the analyzed magnesium alloy were carried out for the temperatures 350, 400 and 450 °C and for the strain rates of 0.1, 1.0 and 10.0 s^−1^. Based on the results of the experimental studies, their approximation used extended the Hensel–Spittel [34] formula for the flow stress *σ_f_* dependence of strain ε, strain rate $\dot{\varepsilon }$ and temperature *T* expressed as:(1)$$\sigma_{f}=A\cdot e^{m_{1}\cdot T}\cdot T^{m_{9}}\cdot \varepsilon^{m_{2}}\cdot e^{\frac{m_{4}}{\varepsilon }}\cdot {\left(1+\varepsilon \right)}^{m_{5}\cdot T}\cdot e^{m_{7}\cdot \varepsilon }\cdot {\dot{\varepsilon }}^{m_{3}}\cdot {\dot{\varepsilon }}^{m_{8}\cdot T},\;\mathrm{MPa}$$

Example flow curves for the AZ31 magnesium alloy under the conditions mentioned above are presented in Figure 1. The solid lines with full markers denote the actual curves (obtained from plastometric tests), while the solid lines with hollow markers denote the curves determined based on approximating with function (1).

The plastometric tests showed that during plastic deformation of AZ31 magnesium alloy at low strain rates, recovery and recrystallization processes take place [35,36,37]. A characteristic decrease in the value of the flow stress is observed after the deformation; a value of approx. 0.4 is exceeded. The maximum values of the flow stress were obtained for the actual deformation not exceeding the value of 0.2.

To increase the accuracy of numerical calculations, the properties of AZ31 alloy are presented in a table format, combining experimental test results (for a specific range) supplemented by the results calculated from Equation (1) using the factors from Table 2. The material model developed by the authors was verified in the research [35,36].

The tests involved using round feedstock with a diameter of 30 mm and a length of 200 mm, made of cast ingots 65 × 120 × 650 mm in size. Before making the feedstock for the rolling process, the ingots were subjected to the homogenization process (annealing for 24 h at 400 °C) to homogenize the structure after casting. For the as-poured material and as-homogenized material, samples were taken for structural tests. Furthermore, for the post-homogenization material, standardized samples for tensile tests were prepared.

Figure 2 shows the initial structure of the as-cast and as-homogenized AZ31 magnesium (as provided by the manufacturer). When analyzing the presented structures, it can be noticed that the tested as-cast alloy has a characteristic casting structure (Figure 2a) with visible inclusions of brittle Mg phases Mg_17_Al_12_ and Mg_17_Al_12_+ α at the boundaries at the dendrite boundaries of a solid solution of aluminum in magnesium [38]. The performed homogenizing annealing results in the dissolution of brittle phases (Figure 2b,c). The observed as-homogenized structure was coarse-grained with a grain size ranging from 150 to 700 μm (the average grain size was 500 μm). When analyzing the structure of the tested alloy, minor inclusions of brittle phases (especially those occurring at the grain boundary) can be observed.

Tensile tests were performed with a Zwick Z100 (ZwickRoell GmbH & Co. KG, Ulm, Germany) testing device, using standard round samples with a diameter of 5 mm for the initial material and the length of the measuring base of 100 mm. However, for as-rolled bars, no samples were taken and the tearing was performed for three ready-made bars with the diameter of 15 mm rolled according to the assumed variants, which was to determine the influence of the obtained bimodal structure on their properties. Furthermore, both for the feedstock and for the samples taken from rolled rods, the microhardness measurement was carried out by the Vickers method using a semi-automatic Future-Tech FM700 (FUTURE-TECH CORP., Kawasaki, Japan) microhardness tester. The microhardness tests were performed for samples taken from rolled bars in accordance with the assumed variant (a sample from each rolled bar), and for each of them, the measurement was taken on the longitudinal section, with the load of 50 g.

Theoretical tests of the rolling process of AZ31 alloy bars were conducted using commercial software, i.e., Forge NxT 2.1 (Transvalor, Biot, France). The visco-plastic deformed metal model and three-dimensional state of strain were assumed for the numerical examination:(2)$$S_{ij}=2K(T,\dot{\bar{\varepsilon }},\bar{\varepsilon }){\left(\sqrt{3}\dot{\bar{\varepsilon }}\right)}^{n-1}{\dot{\varepsilon }}_{ij}$$
where: *S_ij_* is the deviatoric stress tensor, $\dot{\bar{\varepsilon }}$ is the equivalent strain rate, *ε_ij_* is the equivalent strain rate tensor, $\bar{\varepsilon }$—equivalent plastic strain, *T* is the temperature, *K*—consistence being dependent on the yield stress σ_p_, *n*—factor characterizing hot metal deformation (0 < *n* < 1).

The friction conditions on the surface of metal-rolls contact are described with the Coulomb friction model and the Tresca friction model, in which appropriate coefficient values are taken:(3)$$\tau_{j}=\mu \cdot \sigma_{n}\;\mathrm{for}\;\mu \cdot \sigma_{n}\leq \frac{\sigma_{0}}{\sqrt{3}}$$
(4)$$\tau_{j}=m\frac{\sigma_{0}}{\sqrt{3}}\;\mathrm{for}\;\mu \cdot \sigma_{n}>m\frac{\sigma_{0}}{\sqrt{3}}$$
where: *τ_j_*—friction stress, *σ*_0_—base stress, *σ_n_*—normal stress, *μ*—friction coefficient, *m*—friction factor.

For the determination of the temperature field, a differential equation is used, which describes the temperature variations for a transient heat flow. This is a quasi-harmonic equation that can be expressed in the following form:(5)$$\frac{\partial }{\partial x}\left(k_{x}\frac{\partial T_{s}}{\partial x}\right)+\frac{\partial }{\partial y}\left(k_{y}\frac{\partial T_{s}}{\partial y}\right)+\frac{\partial }{\partial z}\left(k_{z}\frac{\partial T_{s}}{\partial z}\right)+\left(Q-c_{p}\rho \frac{\partial T_{s}}{\partial t}\right)=0$$

In the above equation, *k_x_*, *k_y_* and *k_z_* are the functions of distribution of thermal conductivity coefficients in the directions *x*, *y*, *z*. The *Ts* is a function used to describe the temperature in the zone under consideration. The *Q* is the function of distribution of the deformation heat generation rate; *c_p_* is the function of distribution of metal specific heat; and *ρ* is the function of distribution of metal density. As the boundary conditions, the combined limiting conditions of the second and third kinds were adopted, which can be written in the form below:(6)$$k_{x}\frac{\partial T_{s}}{\partial x}l_{x}+k_{y}\frac{\partial T_{s}}{\partial y}l_{y}+k_{z}\frac{\partial T_{s}}{\partial z}l_{z}+q+\alpha_{k}\left(T_{s}-T_{ambient}\right)=0$$

In the above equation, *l_x_*, *l_y_* and *l_z_* are the directional cosines of the normal to the band surface, *q* is the intensity of heat flow over the cooled zone surface and *α_k_* represents convective losses. Equation (5) and boundary condition (6) uniquely define the heat exchange during modeling of the rolling process.

The die model was constructed from a triangle-based mesh, while the AZ31 feedstock model was constructed from tetrahedral elements. During deformation, the volumetric mesh is automatically remeshed, depending on the deformation value in order to avoid excessive deformation and improving the quality of the solutions. The AZ31 feedstock model was built from ca. 120 thousand elements (each element of average edge length was different for each rolling pass from 1.0 for the first pass to 0.6 in the last pass), and ca. 20 thousand elements for working rolls, each element having average edge length of 2 mm.

The above software enables the modeling of complex deformation processes, and with the use of a cylindrical coordinate system, it can reproduce the RSR process. The theoretical analysis was performed for the real rolling conditions: rotational speed—100 rpm, friction factor—0.8, coefficient of heat exchange between the material and the tool α—5000 W/(K·m^2^); coefficient of heat exchange between the material and the air α_air_—10 W/(K·m^2^), working rolls temperature—30 °C; ambient temperature—20 °C. To introduce a computer model, a model of the RSR SVP-08 laboratory rolling mill (ZAO “ISTOK ML”, Moscow, Russia) was used, located at the Czestochowa University of Technology (Częstochowa, Poland) (Figure 3). In the three-high skew rolling mill used for the rolling process, the roll axis inclination angle is 18°, while for conventional radial shear rolling mills, which are used to the tubes and bars production, the maximum axis inclined angle is 16°.

The pattern of deformations used in the tests for the analyzed variants of the RSR process on AZ31 alloy bars is presented in Table 3. The tests were carried out in two sets of work rolls with respective diameters of 90 mm and 73 mm (due to the design limitations of the RSR SVP-08 rolling mill). Bars with diameters ranging from 30 mm to 20 mm were rolled in a set of rolls with a diameter of 90 mm, while bars ranging from 20 mm to 15 mm were rolled in rolls with a diameter of 73 mm.

In order to assess the impact of changing the degree of the unit deformation on the possibility of a gradient structure, the sum of deformations for subsequent passes was analyzed.

Before each rolling pass, the samples were heated in an electric chamber furnace LAC KC 120/14 (LAC, Židlochovice, Czech Republic) to the assumed initial rolling temperature of 400 °C.

## 3. Results and Discussion

### 3.1. Theoretical Analysis of the RSR Process for AZ31 Magnesium Alloy Bars

Due to the nature of the deformation in the RSR rolling process, the parameter to identify the possibility of the formation of a bimodal structure in the rolled bar is the effective deformation, defined as the sum of deformations in subsequent passes in each analyzed rolling variant (Figure 4).

By analyzing the computer simulations presented in Figure 4, it can be seen that the obtained distributions in the transverse planes are of a radial nature. It can be noted that for Variant I, the difference in the values for subsequent zones (from the bar axis to the surface) have a lower gradient than for Variant III. Therefore, it can be presumed that the observed radial distribution will result in a differentiation of the structure on the cross-section of the finished products. For all analyzed variants in the axial zone of the rolled bar, the lowest effective deformation values are observed and they are respectively: 13.8–14.6 for Variant I, 13.4–14.5 for Variant II and 13.2–14.2 for Variant III. The differences in the values of effective deformation may result from the fact that despite the use of smaller values of deformation in a single pass for Variant I, increasing the number of passes results in an increase in the value of deformations associated with the torsion of the rolled material (shear strain towards the direction of twisting *γ_ρz_*) in the axial zone. On the other hand, the use of a larger unit deformation in a single pass with a smaller number of passes will result in an increase in the proportion of shear strain in the direction of twisting *γ_ρz_* in the near-surface zone.

The changes in the nature of the effective strain distributions for the analyzed variants result from the differences in the velocities and directions of metal flow (Figure 5) during the RSR process.

Figure 6 presents the distribution of stress tensor component *τ_ρz_* on the longitudinal section (deformation zone) of the rolled bar and on the cross-section (located in the exit of the deformation zone) for intermediate passes in each rolling variant (characterized by a different value of deformation).

When analyzing the data presented in Figure 6 (it is assumed that a positive value of tangential stresses means that they try to “rotate” the single-unit element clockwise), it can be noticed that increasing the unit deformation does not result in a significant change in the value of stress tensor component *τ_ρz_*; however, when analyzing the obtained distributions for cross-sections, the value of the stress tensor component *τ_ρz_* increases with the increase of the deformation in a single pass at the place of contact of the deformed metal with the working tools. Furthermore, in a larger volume of the deformed metal, the analyzed tensor component changes the orientation.

Additionally, an analysis of changes in the distribution of shear strain in the direction of torsion *γ_ρz_* was performed for intermediate passes in each of the rolling variants (characterized by a different level of deformation), Figure 7.

The analysis was performed for pass 3 (diameter reduction from 24 mm to 21 mm, 1.3 elongation) for Variant I, pass 3 (diameter reduction from 24 mm to 20 mm, 1.44 elongation) for Variant II and pass 2 (diameter reduction from 25 mm on 20 mm, 1.56 elongation) for Variant III. The distribution of shear strain presented in Figure 7 confirms that the distribution of strain tensor component *γ_ρz_* on the cross-section is sequential (symmetrical with respect to the axis of symmetry of the rolled bar) associated with the torsion and change in the length of the spiral line along which the deformed metal flows [25,33,39]. The observed change in the sign of the values of the shear strains only determines the direction of calculating the value of deformation tensor component *γ_ρz_*. Along with the increase of the level of deformation, a decrease in the longitudinal length of the range can be observed, where the analyzed component has a positive direction, from approx. 25.7 mm for the smallest deformation (Figure 7a) to approx. 20.3 mm for the largest deformation (Figure 7c). Increasing the deformation in individual passes results in an increase in the value of strain tensor component *γ_ρz_* in the near-surface zones (from 0.2 for Variant I to 0.4 for Variant II). It can, therefore, be assumed that the use of a larger number of passes with a lower deformation in a single pass has a more beneficial effect on the homogenization of deformations between the surface and the intermediate and central zones (Figure 4). On the other hand, the use of a smaller number of passes with more significant deformation in a single pass increases the non-uniformity of deformations between the analyzed zones. The local zones of maximum shear strain values *γ_ρz_* observed in near-surface layers (in the range of absolute values from 0.2) are a natural site of initiation of the formation of local shear bands. As previous research in this field has shown, there is significant fragmentation of the structure [31,33], which results from the activation of the shear bands, regardless of the value of the set deformation in a single pass in the surface zones, thus resulting in an increase in the value of shear strain *γ_ρz_* (Figure 7).

In the axis of the rolled bar, the value of the component of shear strain *γ_ρz_* for all analyzed cases does not exceed 0.1. Increasing the value of shear strains *γ_ρz_* is closely related to the reduction of the length of the zone and to the diameter at which they have a constant sign (positive or negative), which results from the reduction of the longitudinal length of twist line in the deformed medium. The obtained distributions and changes in the value of the component of shear strains *γ_ρz_* translates directly into the decomposition of the effective deformation, where in the variant with a greater number of passes with a relatively small unit deformation (Variant I), the effective deformation values in the bar axis are higher compared to the application for rolling of a smaller number of passes with a higher deformation in a single pass (Variant II).

### 3.2. Experimental Analysis of the RSR Process for AZ31 Magnesium Alloy Bars

To determine the effect of the applied rolling variants on the mechanical properties of the finished AZ31 alloy bars, the mean values of the yield strength (YS) and the ultimate tensile strength (UTS) were determined in the tensile test. These results were compared to the value of the tested as-homogenized alloy in Figure 8a. When analyzing the obtained results, it can be concluded that the application of the RSR rolling process results in an increase in both YS and UTS. The highest values of both parameters were obtained for bars rolled according to Variant I, with the smallest for bars rolled according to Variant III. The difference was about 12% and may result from the greater heterogeneity of the deformation distribution on the cross-section of the rolled bars according to Variant III (Figure 4c). Figure 8b shows a photo of fractures at the location of the cracking. It can be observed that the fracture is brittle for both as-rolled bars. Additionally, it follows the helix of the torsion that is forced in the rolling process, which for Variant I is elongated (due to the use of small deformation in a single pass) compared to Variant III (large deformation in a single pass).

Based on the tests of the microhardness of the studied as-homogenized AZ31 alloy, the average determined value was HV_0.05_ = 37. On the other hand, the determined average microhardness (the average of 6 measurements, two for each rolled sample in the assumed rolling variant) values determined for rolled bars on the longitudinal section as a function of the distance from the axis of the rolled bar are shown in Figure 9. When analyzing the data presented in Figure 9, it can be noticed that for all analyzed variants of rolling, the highest values of microhardness are observed for the outer zones of bars (zones of the highest total and shear strains). Then, as the bar axis approaches (in the transition zone), the microhardness decreases and remains at a similar level in the central zone. The smallest differences in microhardness for the cross-section are observed for Variant I and the largest for Variant III. The microhardness values indicated confirm the differentiation of properties for RSR-rolled bars, for which an increased hardness is obtained for the radial outer zone and a reduction in hardness in the intermediate and central zones, which was also confirmed for other materials [17,25]. Increasing the microhardness in the near-surface zone is related to its grain fragmentation, which is caused by the impact of shear stresses and the appearance of additional shear strains.

Figure 10, Figure 11 and Figure 12 present images of the microstructure showing the outer, middle and central zones for AZ31 alloy bars rolled according to the adopted variants. The obtained structure is heterogeneous, with a highly fragmented structure in the surface layers and significantly larger grains in the remaining part. The structure shows twin and recrystallized grains of various orientations (specifically, in the central zones and in a smaller number of intermediate zones). A list of the determined grain sizes for the analyzed rolling variants is presented in Table 4. When analyzing the data showing the microstructure in the surface zones for all passes, a strong fragmentation of the structure is observed, whereas in the case of Variants II and III, clusters of larger partially recrystallized grains can be observed, which occur in a smaller number for Variant I. This may result indirectly from the values of strain tensor component *γ_ρz_* observed in subsequent passes, which for Variants II and III have larger local maxima that contribute to the fragmentation of the structure. Nevertheless, reducing the number of passes causes a higher heterogeneity in the structure in the final product. On the other hand, for Variant I, the greater number of passes (despite lower values of local maxima of strain tensor component *γ_ρz_*) has a positive effect on the uniformity of the grain size. When analyzing the data showing the microstructure in surface zones, a strong fragmentation of the structure is observed for all passes, while in the case of Variants II and III, clusters of larger partially recrystallized grains can be observed, which are smaller for Variant I. This may result indirectly from the values of strain tensor component *γ_ρz_* observed in subsequent passes, which for Variants II and III have larger local maxima that contribute to the fragmentation of the structure.

In the transition zones, an increased number share of larger grains with colonies of a finely divided structure can be observed for Variant I. The obtained structure evidences the formation of shear bands in the RSR process, in which grain refinement occurs in the surface zone through activating additional deformation mechanisms in the AZ31 magnesium alloy. For Variants II and III, a decrease in the number of small, finely divided grains is observed, while the maximum size of single magnesium grains increases. This tendency continues for the central zone in all analyzed variants. The structure becomes coarser with an increased amount of distinct twins and recrystallized grains. In the central zone, as shown in theoretical tests, the value of the set deformation in the subsequent passes is smaller than the intermediate and outer zones, which, at a given rolling temperature, favors recrystallization.

The determined average grain sizes for all variants evidence that as a result of RSR process, it is possible to obtain an uneven distribution of magnesium grain sizes (from the smallest in the near-surface zone to the largest in the central zone), which proves its bimodality. The results obtained for theoretical and experimental tests confirm the possibility of using the RSR process for the production of bars with a bimodal structure from magnesium alloys. Despite the possibility of generating relatively large deformations in subsequent single passes on the RSR rolling mill, the results of the tests carried out have evidenced that it is the most advantageous for the tested alloy to conduct the process with a greater number of passes with smaller single deformation than the use of larger unit deformations and reducing the number of passes. Moreover, due to the fact that for the central zones of the bar rolled in the RSR technology, the deformation is smaller than in the outer zones. It seems beneficial to conduct preliminary plastic working of the feedstock, which is characterized by high deformation in the central part of the processed bar. For this purpose, the process or rolling in modified stretching passes can be used, which provides the fragmentation of the structure in the central zone of the manufactured bars [40].

## 4. Conclusions

The theoretical and experimental tests carried out have evidenced that the use of various deformation patterns affects the obtained structure and grain refinement. Based on the conducted research for the rolling process at 400 °C, it can be concluded that the use of a larger number of passes with a smaller cross-section reduction will positively impact the formation of a bimodal structure consisting of a highly fragmented near-surface structure and in the half of the radius of the structure of fragmented grains at the boundaries of larger grains. Strong grain fragmentation is the result of shear bands that constitute locations of nucleation and dynamic recrystallization of grains, which is beneficial to the fragmentation of the structure. When larger deformations are used during rolled bars, larger grains with distinct twins are visible. In the tested variants of rolling in the twinning areas, no formation of new embryos and no dynamic recrystallization were observed. The use of a smaller number of passes does not ensure complete fragmentation of the structure in the near-surface layers of the bar and results in a band structure consisting of larger grains surrounded by smaller grains. The grain refinement in the areas near the axial of the bar is characterized by a larger grain size due to the small impact of shear strain. Elongation of grains can be expected in areas near the axial zones due to the occurrence of only tensile stresses resulting from the elongation of the samples.

## Figures and Tables

**Figure 1 materials-15-00954-f001:**
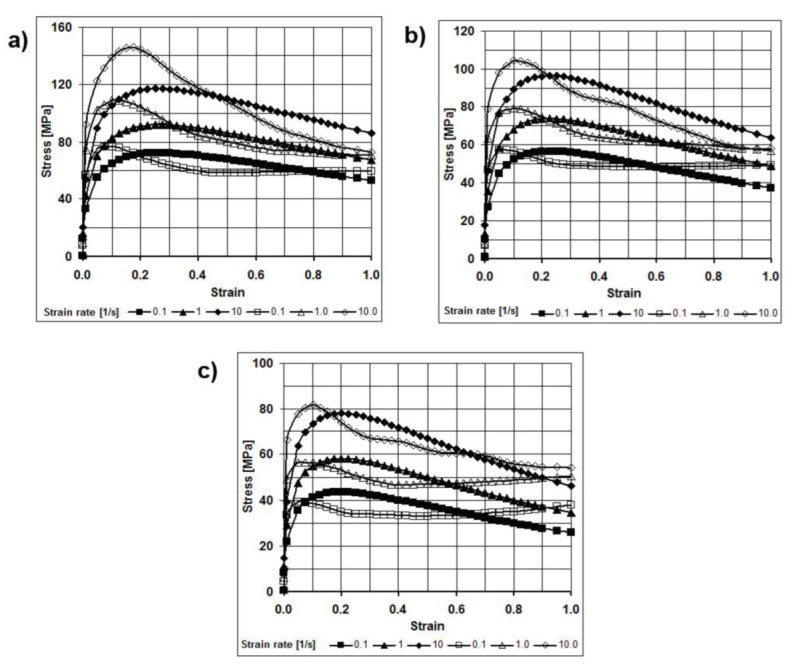
Plastic flow curves for the AZ31 magnesium alloy for temperatures: (**a**) 350 °C, (**b**) 400 °C, (**c**) 450 °C [33].

**Figure 2 materials-15-00954-f002:**
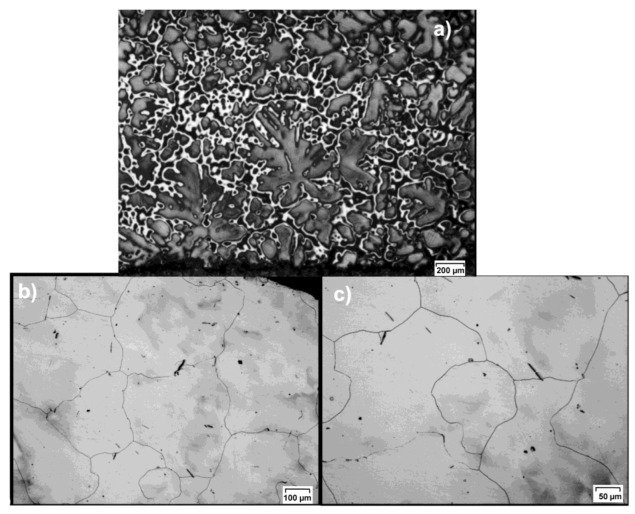
The structure of the AZ31 magnesium alloy used in the tests: (**a**) as-cast (provided by the manufacturer), mag. ×50, (**b**) s-homogenized, mag. ×100, (**c**) mag. ×200.

**Figure 3 materials-15-00954-f003:**
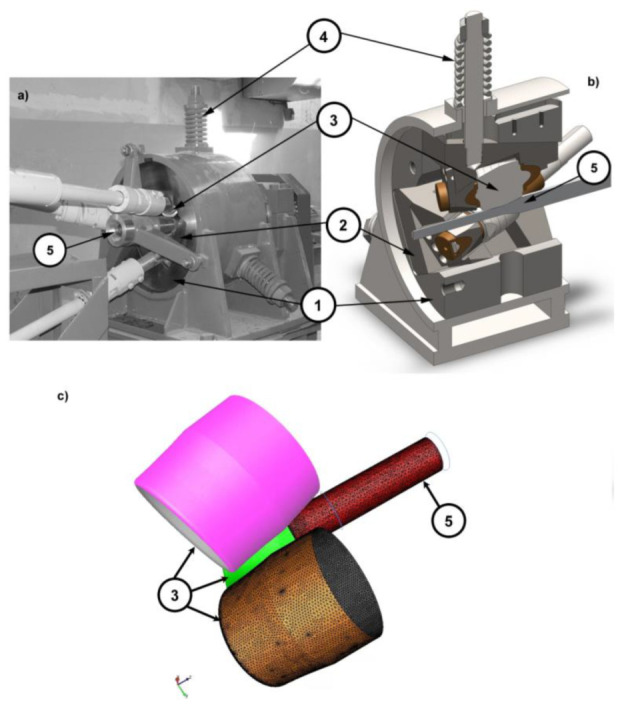
View of RSR SVP-08 rolling mill (Czestochowa University of Technology, Częstochowa, Poland) used for experimental research (**a**); and its assembly, a CAD model (**b**); assembly of FEM elements used for theoretical research in the Forge NxT 2.1 computer program (**c**): 1—rolling mill body, 2—a roller insert with a mounting for work rolls, 3—work rollers, 4—binding screw, 5—rolled stock.

**Figure 4 materials-15-00954-f004:**
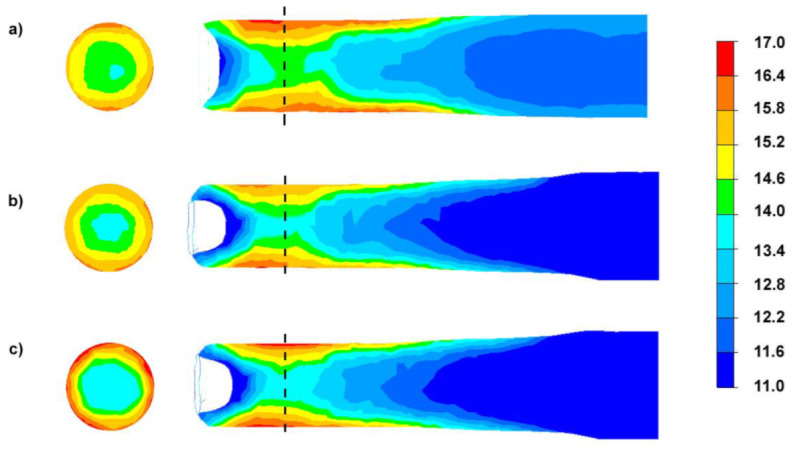
The effective strain distribution for the RSR process of bars from AZ31 alloy: (**a**) Variant I, (**b**) Variant II, (**c**) Variant III.

**Figure 5 materials-15-00954-f005:**
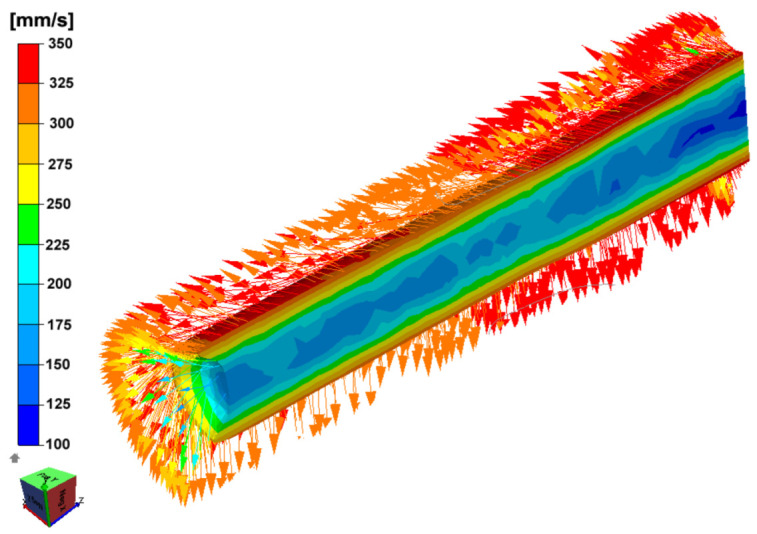
Distribution of metal flow velocity and vectors flow directions for the rolling process of AZ31 alloy bars (Variant I-pass no. 6).

**Figure 6 materials-15-00954-f006:**
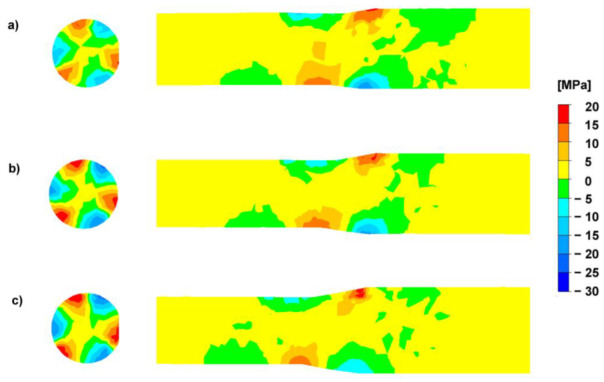
Distribution of the stress tensor component *τ_ρz_* for the RSR rolling process of AZ31 magnesium alloy bars: (**a**) Variant I, pass 3, (**b**) Variant II, pass 3, (**c**) Variant III, pass 2.

**Figure 7 materials-15-00954-f007:**
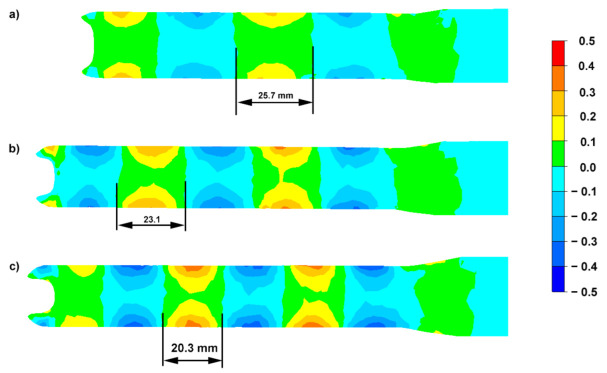
Distribution of the shear strain component *γ_ρz_* for the RSR process of AZ31 magnesium alloy bars: (**a**) Variant I, pass 3, (**b**) Variant II, pass 3, (**c**) Variant III, pass 2.

**Figure 8 materials-15-00954-f008:**
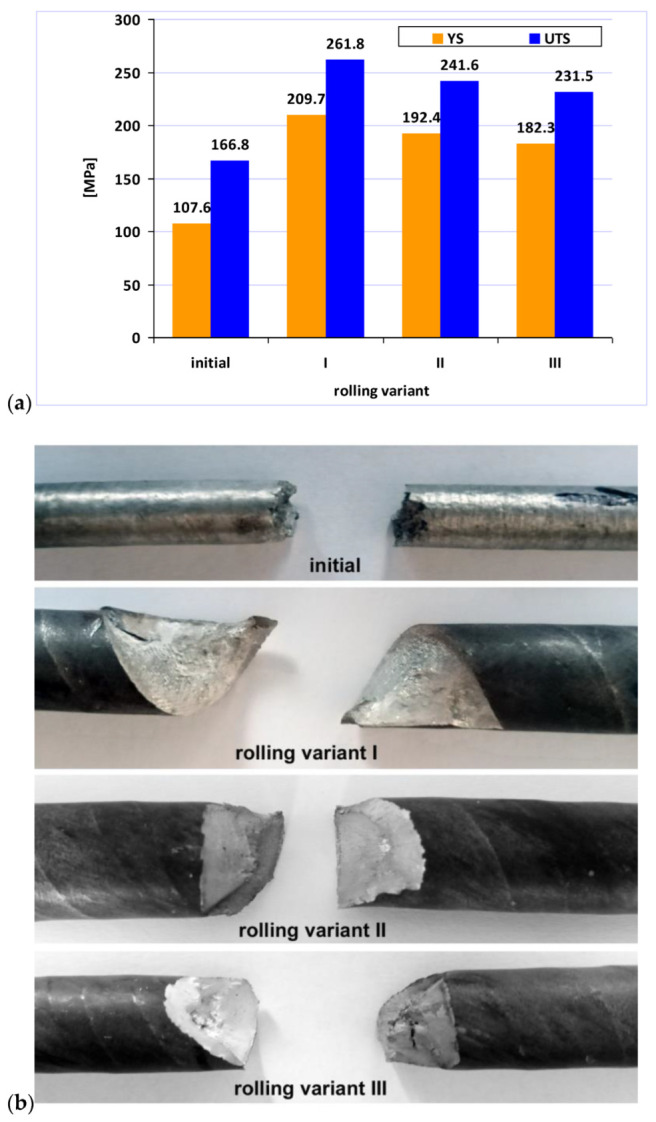
Mechanical properties of AZ31 magnesium alloy in the initial state and after the RSR process; (**a**) view of fractures in the samples after the tensile tests (**b**).

**Figure 9 materials-15-00954-f009:**
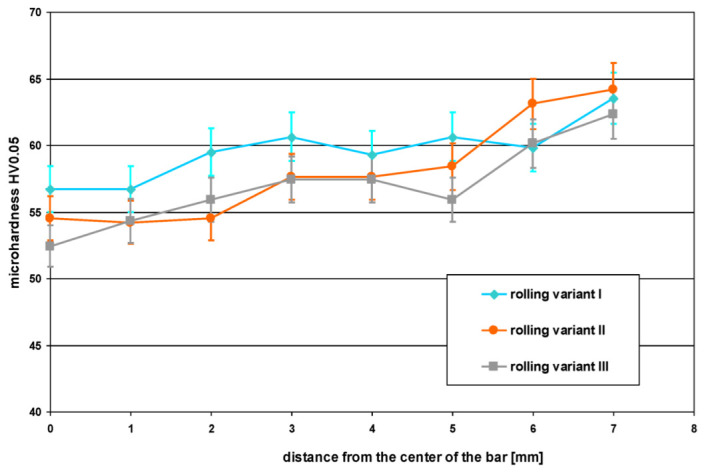
The change of microhardness HV_0.05_ determined on the longitudinal section of rolled bars as a function of the distance from the axis of the rolled bar.

**Figure 10 materials-15-00954-f010:**
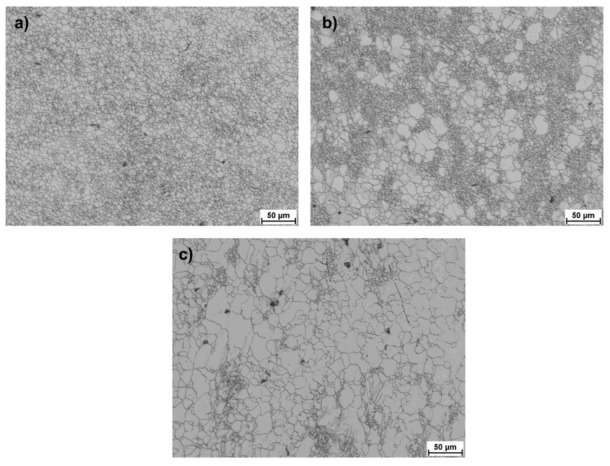
Microstructure obtained for finished bars with the diameter of 15 mm rolled according to Variant I, defined for (**a**) surface zone, (**b**) middle zone, (**c**) central zone.

**Figure 11 materials-15-00954-f011:**
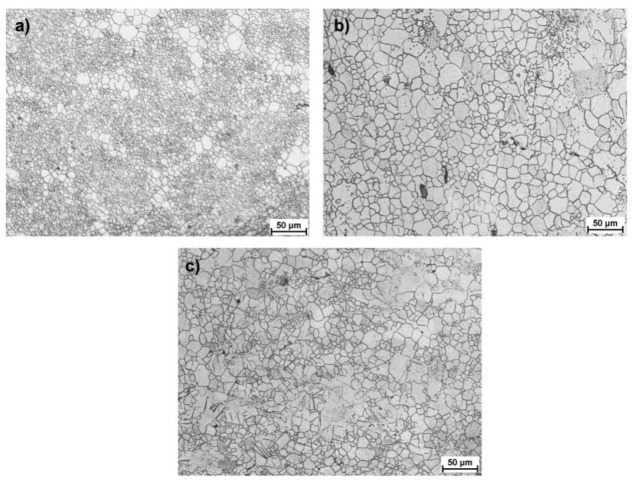
Microstructure obtained for finished bars with the diameter of 15 mm rolled according to Variant II, defined for (**a**) surface zone, (**b**) middle zone, (**c**) central zone.

**Figure 12 materials-15-00954-f012:**
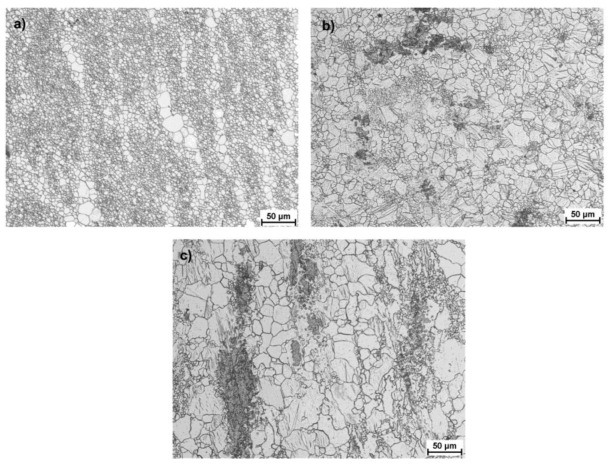
Microstructure obtained for finished bars with the diameter of 15 mm rolled according to Variant III, defined for (**a**) surface zone, (**b**) middle zone, (**c**) central zone.

**Table 1 materials-15-00954-t001:** Chemical composition of the AZ31material used for the tests.

Chemical Composition, % Mass						
Mg	Mn	Cu	Zn	Al	Si	Fe
balance	0.47	0.05	0.99	3.5	0.1	0.01

**Table 2 materials-15-00954-t002:** Parameters of function (1) for the AZ31 magnesium alloy [33].

A	m_1_	m_2_	m_3_	m_4_	m_5_	m_7_	m_8_	m_9_
0.684788	−0.00721633	0.342418	0.02864	0.000230534	−0.00439388	−0.08198	0.0002181	1.41094

**Table 3 materials-15-00954-t003:** The pattern of deformations used in the RSR process for individual variants of rolling.

Rolling Variants	Initial Diameter 30 mm	Diameter after Pass					
		Rolls Diameter 90 mm			Rolls Diameter 73 mm		
		Pass 1	Pass 2	Pass 3	Pass 4	Pass 5	Pass 6
I	diameter (mm)	27.00	24.00	21.00	19.00	17.00	15.00
	elongation	1.23	1.26	1.30	1.22	1.24	1.28
	deformation	0.100	0.111	0.125	0.095	0.105	0.118
	cross-section reduction (%)	19.0	21.0	23.4	18.1	19.9	22.1
II	diameter (mm)	27.00	24.00	20.00			15.00
	elongation	1.23	1.27	1.44			1.78
	deformation	0.100	0.111	0.167			0.250
	cross-section reduction (%)	19.0	21.0	30.6			43.8
III	diameter (mm)	25.00	20.00				15.00
	Elongation	1.44	1.56				1.78
	Deformation	0.167	0.200				0.250
	cross-section reduction (%)	30.6	36.0				43.8

**Table 4 materials-15-00954-t004:** Grain sizes for the analyzed variants of the RSR process.

		Minimum, μm	Maximum, μm	Average, μm
**Initial**		150	800	500
Variant I	surface zone	2	12	4
	middle zone	2	35	18
	central zone	4	63	37
Variant II	surface zone	2	38	5
	middle zone	2	43	26
	central zone	4	52	41
Variant III	surface zone	2	41	5
	middle zone	3	49	27
	central zone	4	63	48

## Data Availability

The data presented in this study are available on request from the corresponding author.

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
