# Peer review of "Changes in the Properties in Bimodal Mg Alloy Bars Obtained for Various Deformation Patterns in the RSR Rolling Process"

_materials, 2022, doi:10.3390/ma15030954_

Round 1
Reviewer 1 Report
This paper is worthy to be published. In order to improve its quality, some comments are provided:
- Grammar mistakes should be avoided, e.g., The works [29]… should be ‘The work [29]…’. Please check the paper throughout.
- Where is No.5 in Fig. 3?
- The meaning of negative or positive of the stress tensors ought to be explained.
- Label of y-axis in Fig. 8 should be provided.
- Elongations of the products should be measured and provided.
- In Fig. 8b, fracture morphology of rolling variant II is forgotten.
- It should be mentioned that the conclusion that multi-pass rolling with smaller reduction of a single pass is better is not general. Temperature may affect this rule. It is better to emphasize your deformation temperature in the abstract and the conclusion.
Author Response
The answers are included in the file

Reviewer 2 Report
Radial shear rolling process (RSR) is a fairly attractive severe plastic deformation (SPD) method to produce bimodal metallic structures. This article has discussed the influences of various RSR deformation paths in depth. However, there are several things need to be clarified before being considered for the publication:
(1) FEM simulation work had been done by the authors. However, nothing about the FEM model has been introduced, except the constitutive model. The information about the FEM model, including the mesh (as well as the mesh convergency test if it is possible), the load, the BCs, the contact conditions, and of course the computational software, should be supplemented.
(2) By the way, the author should explain how were the constitutive parameters in the extended Hensel-Spittel model identified?
(3) The results of the micro hardness tests in Figure 9 should have error bars, and how many testing points had been repeated should be explained as well.
(4) From the results of the micro hardness tests, the author claimed that “Increasing the microhardness in the near-surface zone is related to the grain fragmentation in it, which is caused by the impact of shear stresses and the appearance of additional shear deformations.”. However, in Reference 30, Akopyan et.al had found the opposite micro hardness gradient which is caused by the dynamic recrystallization of the surface fine grain structure. The target material of Reference 30 was Al-Zn-Mg-Cu alloy and the RSR temperature was 480℃. For Al-Zn-Mg-Cu alloy, its recrystallization temperature should be more or less 430~480℃. So, the dynamic recrystallization of the surface fine grain structure occurs naturally. The target material of this work was AZ31 magnesium alloy, the recrystallization temperature of which should be something like 150~200℃. However, the RSR temperature of this work was 400℃, which is much higher than the corresponding recrystallization temperature of AZ31. But why recrystallization did not happen in the surface fine grain structure? How to explain such a phenomenon?
(5) The author claimed that the load of the hardness was 50g. Thus, the unit of the hardness values should be HV0.05 but not HV0.5.
(6) In the legend of Figure 9, it should be “rolling” but not “rooling”.
(7) The poor English expression of the whole manuscript seriously affects its readability and it should be significantly improved before being considered for the publication:

Author Response
The answers are included in the file

Round 2
Reviewer 2 Report
(1) The way the authors reply the 1st comment is fairly bad. The reviewer has never ever seen any numerical simulation researchers describe their FEM model in such a rough and hasty way, for instance, don’t even show the mesh. It cannot be accepted if the authors insist on not improving the description of the FEM model.
(2) When replying the 2nd comment, the authors mentioned that “for strains in the range 0.1÷2 and strain rates from 0.1 to 10.” The reviewer has never ever seen any researchers describe strain in a manner that “0.1÷2”, and describe strain rate without any unit.
(3) The authors have corrected HV to HV0.05 in the text, but have not corrected the corresponding unit in Fig.9.
(4) Last but not least, the reviewer has been shocked by the reply of the 7th comment. This is the first time that the reviewer has seen a scientific research article “translated by a translator”.
Overall the research work of this article is not bad. However, the way the authors describe their work and reply the comments is quite bad. Hope they can make further improvements in organizing their work and replies to make their research work worthwhile.
Author Response
The answers are included in the file.